# CR-Guided Transformers: Coherence-based Redundancy Identification and Regularization

## Abstract

Current Transformer-based language models demonstrate excellent performance across various tasks. However, these models commonly produce redundant transformations in middle-to-deep layers. This manifests as transformations between the inputs and output in a layer containing pronounced linear correlation or near irrelevance components. This paper attributes its root cause to current training paradigms. These paradigms emphasize prediction accuracy while neglect the effectiveness of nonlinear transformations in model layers. Based on this observation, we propose criteria for identifying redundant transformations. To quantify the degree of redundancy, we further propose a Coherence-based Redundancy (CR) measure. Specifically, we treat the input and output of a model layer as sequence distributions. We leverage characteristic functions and Fourier transform to map the distributions to frequency-domain representations. Finally, we compute coherence in the complex plane and assess the effectiveness of transformations on a [0,1] coherence scale. To suppress redundant transformations at layer outputs, we propose two schemes: tree-structured residual paths and a coherence-based redundancy loss. These approaches guide middle-to-deep layers to produce effective transformations. At the same time, they supervise and regularize against redundant outputs. Our pre-training experiments on Llama3-130M with 12 layers demonstrate that the proposed methods significantly reduce redundant transformations. With training settings held constant, we successfully make the 12-layer model outperform the 14-layer baseline.

## 1 Introduction

Current large language models (LLMs) are predominantly built on the Transformer architecture Vaswani et al. (2017); Devlin et al. (2019). They have achieved remarkable success in intelligent question answering, machine translation, code generation, and so all. LLMs typically employ the Pre-Norm approach to make training stabilize Xiong et al. (2020); Liu et al. (2020b). They also use residual connections He et al. (2016) to mitigate gradient vanishing in deep networks. Despite the impressive performance, prior researches Sun et al. (2025); Men et al. (2024) indicate that Transformer models with Pre-Norm suffer from increasingly similar hidden features in deeper layers. This is a phenomenon known as representation collapse Liu et al. (2020a). Moreover, this issue is widespread across mainstream LLMs that employ the Pre-Norm configuration Sun et al. (2025).

Motivated by this phenomenon, many works perform compression Zhu et al. (2024b); Yin et al. and pruning Frantar & Alistarh (2023); Ma et al. (2023) on the middle-to-deep layers of LLMs. The results show that replacing Lad et al. (2024) or removing Men et al. (2024) portions of the middle-to-deep layers barely affects the performance of LLMs. Notably, CoD Sun et al. (2025) experimentally demonstrates that "more than half of the layers in LLaMA2-13B Touvron et al. (2023b) can be safely removed". These studies effectively reveal strong robustness of LLMs to pruning or replacing operations applied to their middle and deep layers. In contrast, pruning the shallow layers significantly degrades model performance. This indicates that the middle-to-deep layers fail to learn unique, effective, and irreplaceable features. The feature-learning efficiency of the middle-to-deep layers is inferior to that of the shallow layers. This implies that current model layers have issues of parameter redundancy or even waste due to insufficient parameter utilization.

However, merely applying pruning or compression to reduce resource waste does not address the root cause of deep-layer representation collapse. Crucially, approaches like pruning Sreenivas et al.; Lu et al. (2024); Siddiqui et al. (2024) cannot ensure that all layers in the original Pre-Norm-based Transformer model are sufficiently trained.

Therefore, we aim to ensure that every layer performs sufficient and effective nonlinear transformations on the input features. In order achieve it, we thoroughly analyze the causes of deep-layer representation collapse and propose effective solutions. We believe that the core reason for deep-layer representation collapse is current LLM training paradigms. They focus primarily on the accuracy of predicting the next token. However, they neglect whether the model layers produce invalid or insufficient nonlinear transformations during feature computation. We call this phenomenon redundant transformations. Based on this analysis and our extensive experimental observations, we establish following operational criterion for identifying redundant transformations. If either of the two conditions below holds, we deem the layer to have produced a redundant transformation of the input features. First, if the input and output features at the same sequence position exhibit pronounced linear correlation, we consider that the layer has not performed a sufficient nonlinear transformation at that position. Second, if the input and output features at the same sequence position are uncorrelated, we consider that the layer has produced invalid features at the corresponding position. Both of cases indicate that the output of layers contains redundant transformations.

To further quantify the degree of redundancy producing during the input-output transformation of a layer, we propose a Coherence-based Redundancy measure (CR). To compute CR, we treat the input and output sequence features of a layer as discrete distributions of high-dimensional random variables over sequence indices. Then, within the theoretical framework of distribution matching Zhao & Bilen (2023), we employ characteristic function Bisgaard & Sasvári (2000) and map the sequence distributions to frequency-domain representations through Fourier transform. On this basis, we compute coherence by structurally measuring differences between distributions in the more separable complex plane. Since the range of coherence values is [0,1], it provides conveniently an indicator to assess the distributional differences and transformation situations between input and output features.

Corresponding to our criterion of redundant transformations, the CR approaching either 1 or 0 signifies the presence of redundancy between the input and output features. Our expectation is to suppress the model layer from producing the aforementioned redundant transformations. This will prompt the model layer to make effective transformations on the input hidden states that are neither simple copies nor irrelevant noise. To achieve this goal, we further propose two effective schemes based on the CR to reduce the redundancy in output of the model layer and improve parameter utilization.

First, by analyzing transformations all model layers with the CR, we observe that middle-to-deep layers perform less sufficient and less effective nonlinear transformations than shallow layers. Guided by this, we design a scheme to enhance cross-layer information flow between shallow and deep layers. Specifically, inspired by dense connections Huang et al. (2017) and Hyper-connections Zhu et al. (2024a), we propose a simpler and more easily implementable tree-structured residual path on top of the original serial residual connection. Through the tree-like residual connection path, shallow-layer features are introduced into deep layers in a skip manner. This not only breaks the cross-layer information bottleneck existing in the serial residual, but also prompts each layer to participate effectively in learning.

Second, based on the CR, we further propose a coherence-based redundancy loss computed along the sequence dimension of hidden states. Through identifying and regularizing redundancy in the hidden states of middle-to-deep layers, we impose penalties on output positions whose coherence approaches 1 or 0. This suppresses redundant layer transformations and prompts the middle-to-deep layers to perform sufficient and effective transformations on the input features. Because the proposed regularization loss is easy to deploy in Transformer architectures, it can be used in parallel with the tree-structured residual path. These schemes together promote each layer of the model to learn rich and effective features.

Additionally, given that redundant transformations are widespread in a series of language model Sun et al. (2025) with the Pre-Norm, this study conducts validation experiments based on a small-scale language model (SLM). We challenge conducting pre-training experiments on redundancy identification and regularization using a model based on the llama3. It has 130M parameters and 12 Trans-

former layers. Without adding parameters, our schemes successfully reduce feature redundancy and significantly enhance the performance of the llama-130M model. Ultimately, with all other training settings held constant, we make 12-layer model based on Llama3 outperform the 14-layer baseline model.

In this study, our contributions are as follows:

- We thoroughly analyze the root causes of the increasing similarity of features in the middle-to-deep layers. And we point out that model layers tend to produce redundant transformations under current LLM training paradigms.

- We elucidate the criteria for identifying redundant transformations. And we further propose the coherence-based redundancy measure (CR) to quantify the degree of redundant transformations produced by model layers.

- Based on the CR, we propose two effective schemes to guide and constrain the middle-to-deep layers. They successfully suppress redundant outputs from model layers and improve feature transformation efficiency.

- We conduct pre-training experiments on redundancy identification and regularization with SLM, which is llama3-130M with 12 Transformer layers. Ultimately, with all other training settings held constant, we enable it to outperform 14-layer baseline model.

## 2 RELATED WORK

Currently, mainstream large language models are mostly based on the Transformer architecture, such as LLaMA Touvron et al. (2023a;b) and the QWen Bai et al. (2023) series of models. These models all employ the Pre-Norm configuration. Pre-Norm refers to applying RMSNorm first, followed sequentially by Self-Attention, FeedForward, and the residual connection, as shown in 1 (b). The specific computational process of the Pre-Norm architecture is as follows:

$$x' = x + \text{Attn}\big(\text{RMSNorm}(x)\big), \qquad y = x' + \text{FFN}\big(\text{RMSNorm}(x')\big).$$

This architecture has been confirmed by numerous studies Xiong et al. (2020); Wang et al. (2024); Takase et al. (2023) to provide stronger training stability and faster convergence for models. However, by analyzing that the average cosine similarity between the input and output features of each layer from the Pre-Norm architecture, we observe an issue. It is very high that the average cosine similarity of some middle-to-deep transformer layers, as shown in 1 (a). According to previous research Men et al. (2024), cosine similarity is often used to measure correlation between features. A high average cosine similarity indicates that these layers perform a little transformation on the input hidden states. In this study, we regard this phenomenon as one indicator of redundant transformations produced by model layers. To provide a mathematical explanation for this phenomenon, we first simplify the Pre-Norm computation to:

$$y = x + F\big(\text{LN}(x)\big).$$

Where $F(\cdot)$ denotes either Attention or FFN, and $LN$ denotes RMSNorm. Next, We analyze the gradient $\dfrac{\partial L}{\partial x}$, where $L$ is the loss.

$$\frac{\partial L}{\partial x} = \frac{\partial L}{\partial y}\frac{\partial y}{\partial x} = \frac{\partial L}{\partial y}\left(\boldsymbol{I} + \frac{\partial F(\text{LN}(x))}{\partial x}\right).$$

From the above equation, for each layer in a Pre-Norm Transformer, the backpropagated gradient contains the identity term $I$ and the residual-branch gradient term $\dfrac{\partial F(\text{LN}(x))}{\partial x}$. If the residual-branch gradient term is too small, the derivative of this layer tends to approach the identity matrix. In that case, the layer performs a little nonlinear transformation on the input hidden states, i.e., an identity transformation Sun et al. (2025). This conclusion is consistent with our observation of high average cosine similarity between the input and output features of the model layers.

Furthermore, we further observe changes in the average cosine similarity between the input and output features of the attention sub-layer. These changes are positively correlated with the corresponding changes between the input and output of the Transformer layer, as shown in 1 (a). And the

average cosine similarity between input and output of the attention sub-layer is higher than that of the Transformer layers. Therefore, if attention sub-layers in some middle-to-deep layers exhibit excessive identity-mapping behavior, the issue propagates throughout the corresponding Transformer layer. This causes a considerable number of neurons in the corresponding model layer to fail to perform nonlinear mappings adequately. As a result, it leads to the model layer producing redundant transformations. When multiple transformer layers produce redundant transformations, it causes a decrease in the diversity of model representations. This weakens the model's learning ability and its capacity to solve complex tasks. Hence, we believe that excessive identity mapping in attention sub-layers is one of the main reasons leading to producing redundant transformations from the model layer.

In addition to the aforementioned approximate identity mapping, we also observe that nearly irrelevant transformations in experiments. Specifically, at certain sequence positions, cosine similarity between the input and output features of the attention sub-layer is significantly low. In this study, we regard this phenomenon as another indicator of redundant transformations produced by model layers. We associate this phenomenon with the "null attention" proposed by previous work Vig & Belinkov (2019). It means that some attention heads focus on tokens lacking semantic content. As a result, the model layer produces invalid transformations on the input features at the corresponding sequence positions. This implies that a portion of neurons in the model layer are not appropriately and effectively activated, which results in considerable parameter redundancy. Therefore, we regard the invalid transformations appearing in the computation process of the attention sub-layers as another cause of redundant transformations, which are produced by the model layers.

Accordingly, this paper is devoted to accurately identifying redundant transformations in layer outputs and to measuring the degree of redundancy produced by each layer. On this basis, we further propose effective schemes to suppress the above redundant transformations in layer outputs. These schemes aim to prompt model layers to perform effective transformations on the input hidden states, which are neither simple copying nor irrelevant noise. Without increasing parameters, these schemes enhance the practically available representational capacity of each layer. Consequently, models of the same scale achieve higher parameter utilization and stronger representational ability.

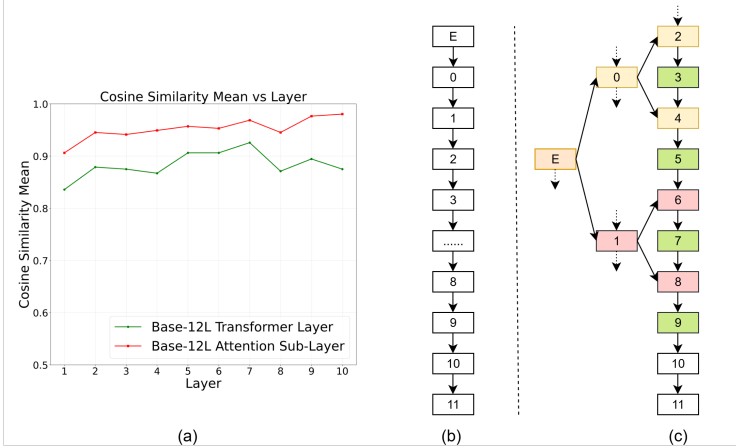

Figure 1: Model architecture and layer similarity metric. (a) shows the mean cosine similarity between input and output of from 1 to 10 Transformer layers and attention sub-layers in the 12-layer baseline. (b) shows a simplified representation of the original serial residual structure of the 12-layer baseline. (c) shows we proposed the tree-structured residual paths based on the original residual connections. Here, E denotes the embedding layer. The dashed arrows indicate the residual paths at the positions corresponding to the (b).

## 3 METHODOLOGY

In this section, we introduce the basic principles and design logic, which are used to identify redundant transformations between layer inputs and outputs and to measure the degree of redundancy.

## 3.1 COHERENCE-BASED REDUNDANCY MEASURE (CR)

According to our observations, some layers in language models contain redundant components, and the degree of redundancy varies across layers. To accurately measure the degree of redundancy between a layer's input and output features, we propose a novel metric called the Coherence-based Redundancy Measure (CR). Although prior research Men et al. (2024) can simply obtain correlations between features by calculating cosine similarity. The metric primarily reflects directional information and remains insensitive to scale changes. And it struggles to capture higher-order statistical differences inherent in nonlinear transformations.

To address this limitation, we treat the input and output sequence features of the model layers as discrete distributions of high-dimensional random variables over the sequence index t. We accurately capture the correlation between high-dimensional features by structurally measuring the differences between the corresponding distributions. For this purpose, within the theoretical framework of distribution matching Zhao & Bilen (2023); Wang et al. (2022), we construct the redundancy metric by leveraging the characteristic function Bisgaard & Sasvári (2000). It is a fundamental and general tool for measuring distributional differences in probability theory. Based on characteristic function, we can map the sequence distribution to a frequency domain representation via the Fourier transform. Moreover, we can measure distribution differences between features and compute coherence within the frequency domain structure. Compared to cosine similarity relying solely on directional information, the difference measure based on characteristic function can utilize richer information. There are complex-valued representations, as well as magnitude and phase in the frequency domain representation. Therefore, frequency-domain coherence can more comprehensively reflect the nonlinear transformation of hidden states. And frequency-domain coherence is more analytical. Combined with the criteria for redundant transformations, frequency-domain coherence of sequence distributions can accurately indicate the degree of redundancy between a layer's input and output. In this paper, we employ the empirical characteristic function in the following form:

$$\phi_x(u) \triangleq \mathbb{E}\Big[e^{\mathrm{i}\,\langle u,x\rangle}\Big] \approx \frac{1}{N}\sum_{n=1}^{N} e^{\mathrm{i}\,\langle u,x_n\rangle}.$$

Specifically, let $h_{\mathrm{in}},\ h_{\mathrm{out}} \in \mathbb{R}^{B\times T\times D}$, where $h_{\mathrm{in}}, h_{\mathrm{out}}$ are the input and output hidden states of the network layer features, and $t = 0..T-1$ is the sequence index. First, we normalize $h_{\mathrm{in}}$ and $h_{\mathrm{out}}$ along the sequence dimension.

$$\widetilde{h}_{\mathrm{in}} = \frac{h_{\mathrm{in}} - \mu_{\mathrm{in}}}{\sigma_{\mathrm{in}} + \epsilon}, \qquad \widetilde{h}_{\mathrm{out}} = \frac{h_{\mathrm{out}} - \mu_{\mathrm{out}}}{\sigma_{\mathrm{out}} + \epsilon}.$$

Where, $\mu$ and $\sigma$ denote the mean and standard deviation along the hidden state sequence dimension, and $\epsilon = 10^{-8}$. On this basis, we apply softmax function along the sequence dimension T to obtain discrete distributions over the index t.

$$p_{\mathrm{in}} = \mathrm{softmax}(\widetilde{h}_{\mathrm{in}}), \qquad p_{\mathrm{out}} = \mathrm{softmax}(\widetilde{h}_{\mathrm{out}}).$$

We set discrete frequency sampling points according to the Nyquist sampling theorem. Using the conjugate symmetry of the DFT, we take only non-negative frequencies $K = \lfloor \frac{T}{2} \rfloor + 1$. And we define orthogonal basis functions.

$$\omega_k = \frac{2\pi k}{T}, \quad k = 0,1,\ldots,K-1; \quad c_{k,t} = \cos(\omega_k t), \quad s_{k,t} = \sin(\omega_k t).$$

Where $k = 0..K-1$ is the frequency index. Next, we compute the empirical characteristic function and obtain the real and imaginary components after orthogonal decomposition.

$$\phi_{\mathrm{in}}(k) = \sum_{t=0}^{T-1} p_{\mathrm{in}}(t)\,e^{-\mathrm{i}\,\omega_k t} = R_{\mathrm{in}}(k) - \mathrm{i}\,I_{\mathrm{in}}(k) = \sum_{t=0}^{T-1} p_{\mathrm{in}}(t)\,(c_{k,t} - \mathrm{i}\,s_{k,t}),$$

$$\phi_{\mathrm{out}}(k) = \sum_{t=0}^{T-1} p_{\mathrm{out}}(t)\,e^{-\mathrm{i}\,\omega_k t} = R_{\mathrm{out}}(k) - \mathrm{i}\,I_{\mathrm{out}}(k) = \sum_{t=0}^{T-1} p_{\mathrm{out}}(t)\,(c_{k,t} - \mathrm{i}\,s_{k,t}).$$

Thus, in the more separable complex plane, we structurally compute coherence between input and output distributions to measure distributional differences and transformations. Specifically, we compute the auto-spectra and cross-spectrum of the two distributions. Scale normalization is performed

on the input and output respectively by calculating the auto-spectra. And the cross-spectrum is computed to quantify the degree of alignment between the input and output in the complex plane. Finally, we compute coherence, which yields stable, comparable, and interpretable values in $[0, 1]$.

$$S_{xx}(k) = \left|\phi_{\text{in}}(k)\right|^2, \qquad S_{yy}(k) = \left|\phi_{\text{out}}(k)\right|^2, \qquad S_{xy}(k) = \phi_{\text{in}}(k)\,\overline{\phi_{\text{out}}(k)},$$

$$\text{Coherence}(k) = \frac{\left|S_{xy}(k)\right|^2}{S_{xx}(k)\,S_{yy}(k)} \in [0, 1].$$

This coherence utilizes the complex-valued information in the frequency-domain structure. It can reveal statistical structural differences after nonlinear transformations more effectively than cosine similarity, which relies solely on directional information. As shown in 1 (a) and 2 (a), in the baseline model the input–output coherence and the cosine similarity exhibit the same trend from 1 to 10 attention sub-layers. This proves the effectiveness of coherence in measuring the transformation between input and output features. Accordingly, we obtain the distributional the Coherence-based Redundancy Measure (CR) constructed from characteristic functions. Since its value range is $[0, 1]$, it can conveniently quantify the difference between the distributions corresponding to the input and output hidden states. Combined with the criteria for redundant transformations, it can be inferred as follows. When coherence approaches 0, it indicates invalid feature transformation. And when coherence approaches 1, it indicates insufficient nonlinear transformation between features. Both cases indicate redundant transformations in the model layer output.

## 3.2 Tree-structured Residual Path

On the basis of quantifying distribution coherence, we analyze the transformation patterns of input and output feature distributions in each attention sub-layer of Transformer models with the Pre-Norm architecture. We observe that in the baseline model, the proportion of coherence for shallow-layer input and output feature distributions falling within the $[0.3, 0.7]$ range is higher, as shown in 2 (b). This demonstrates that shallow layers perform more sufficient nonlinear transformations compared to deep layers. To make deep-layer transformations more sufficient and reduce redundant outputs from deep layers, we first modify the residual paths. We aim to strengthen cross-layer communication between deep and shallow layers. According to prior research, dense connection structures Huang et al. (2017); Xiao et al. (2025) serve as a promising solution to enhance information flow cross layer in deep networks. In addition, Hyper-connections Zhu et al. (2024a) reports a $\Lambda$-shaped connection pattern, where shallow layers (e.g., layers 1 and 2) are frequently reused by many subsequent layers. Inspired by these studies, based on the original serial residual connections, we propose a tree-structured residual connection path. It is simpler and easier to implement than Hyper-connections. This path directly feeds the output features of the layer 0 and 1 in the shallow layers into the deeper layers according to a binary tree structure.

Specifically, referring to the binary tree configuration, we fix the embedding layer as the root node. And we fix transformer layer 0 and layer 1 as the two child nodes of the embedding layer. Finally, we continue to select nodes from the remaining layers of the model to construct a full binary tree. In this paper, we take Llama3-130M as the research object. This model contains an embedding layer and 12 transformer layers numbered from 0 to 11, as shown in 1 (b). The 12 Transformer layers can accommodate at most a full binary tree of height 3 containing 7 nodes, according to the following formula: $N = 2^h - 1$. Where $N$ is the number of nodes in the full binary tree, and $h$ is the height of the binary tree. Accordingly, from layers 2 to 10 we select leaf nodes as the child nodes of layers 0 and 1 respectively. Based on this, we can inject outputs of layers 0 and 1 into deep layers via the tree-shaped residual path. Specifically, we choose even-indexed layers 2 and 4 as child nodes of layer 0, and layers 6 and 8 as child nodes of layer 1, as shown in 1 (c). At the same time, we select odd-indexed layers 3, 5, 7, and 9 as buffer layers between the leaf nodes of the tree structure. This aims to reduce conflicts during information interaction between shallow and deep layers and mitigate fluctuations in deep layer features during learning.

By adjusting the original residuals with our tree-structured residual path, we make nonlinear transformations become more sufficient in each model layer. As shown in 2 (a), in the model with the tree-structured residual path, the coherence of each attention sub-layer is significantly lower than that of the baseline. And compared to the baseline, the proportion of coherence for input and output feature distributions with tree-structured residual path falling within the range $[0.3, 0.7]$ is significantly higher, as shown in 2 (b) and (c). This result indicates that, guided by coherence-based

redundancy analysis, the tree-structured residual path adjustment makes each model layer produce fewer redundant transformations.

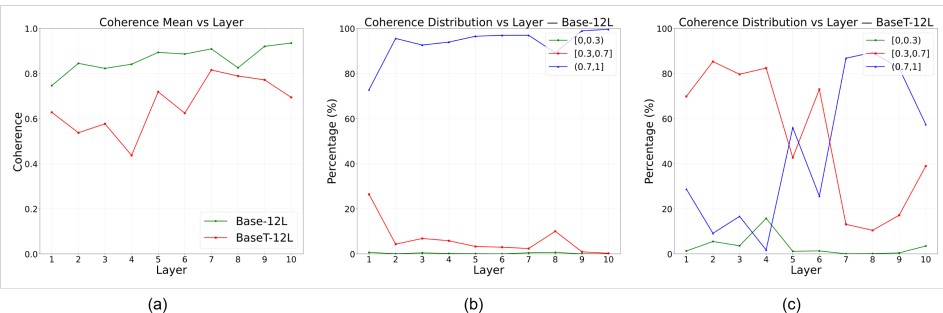

Figure 2: Comparison of CR with different residual paths for attention sub-layers 1-10. (a) shows the coherence of attention sub-layers 1-10 for both the baseline and the model adjusted with the tree-structured residual path. (b) shows the percentage distribution of coherence values in different ranges for attention sub-layers 1-10 in the baseline model (Base-12L). (c) shows the percentage distribution of coherence values in different ranges for attention sub-layers 1-10 in the model adjusted with the tree-structured residual path (BaseT-12L).

## 3.3 REDUNDANCY REGULARIZATION

To further reduce redundant transformations produced by model layers, we design a coherence-based redundancy loss on top of the coherence-based redundancy measure (CR). This allows us to supervise whether the model layers are producing redundant transformations and apply corresponding regularization constraints.

With the same computation process as CR, we let $h_{\text{in}},\ h_{\text{out}} \in \mathbb{R}^{B \times T \times D}$, where $h_{\text{in}},\ h_{\text{out}}$ are the input and output hidden states of the network layer features. Since different layers exhibit different variances and different ranges of values, we aim to bring the feature values of each layer into an appropriate range. Therefore, after normalizing $h_{\text{in}}$ and $h_{\text{out}}$ along the sequence dimension, we first enforce an upper bound on the standardized features. Second, we scale them with a sharpening factor to adjust the sharpness of the discrete distribution by softmax processing.

$$\|\widetilde{h}_{\text{in}}\|_{\infty} \leq b_L = 2 + \frac{L}{2}, \qquad \text{factor}_{\text{in}} = \sqrt{12 - L}, \qquad p_{\text{in}} = \text{softmax}\big(\widetilde{h}_{\text{in}} \times \text{factor}_{\text{in}}\big),$$

$$\|\widetilde{h}_{\text{out}}\|_{\infty} \leq b_L = 2 + \frac{L}{2}, \qquad \text{factor}_{\text{out}} = \sqrt{12 - L}, \qquad p_{\text{out}} = \text{softmax}\big(\widetilde{h}_{\text{out}} \times \text{factor}_{\text{out}}\big).$$

Here L is the layer index. A larger index indicates a deeper position in the model. To apply different regularization strengths to different layers, we set the scaling factor based on the layer index. To prevent excessive fluctuations during training in middle-to-deep layers, we make the scaling factor decrease as the layer index increases. This can make the discrete distributions corresponding to deeper layer features smoother. Subsequently, following the same computation process as CR, we set discrete frequency samples and use the empirical characteristic function to map sequence distributions into the complex domain of frequency. After obtaining coherence based on the cross-spectrum and auto-spectra of the two distributions, we further compute the Coherence-based Redundancy loss as follows.

$$\mathcal{L}_{CR} = (Coherence - target)^2 \times scale_L, \qquad scale_L = (12 - L) \times 0.1.$$

Here, we set corresponding targets based on the redundancy situations in different model layers. And we penalize outputs where coherence approaches 1 or 0, thereby suppressing redundant transformations produced by the model layer. Meanwhile, to prevent large training fluctuations in middle-to-deep layers caused by the CR loss, we make $scale_L$ decrease as the layer index increases. This aims to minimize redundant transformations while improving next-token prediction accuracy.

To further constrain redundant transformations, in addition to supervising the outputs along the sequence dimension with the CR loss, we also apply an explicit constraint on them along the channel

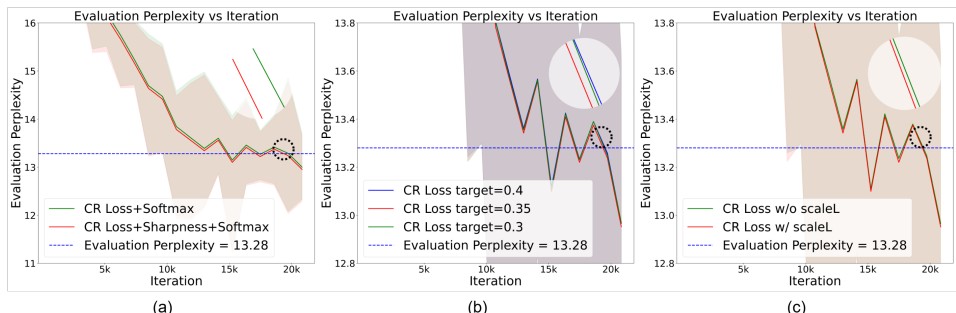

Figure 3: Ablation study. To make the comparison of evaluation perplexity among models with different hyperparameters more distinct, the curves within the dashed circle are magnified by a factor of 2.5. And they are displayed in the upper-right region of the line chart. The best evaluation results are consistently plotted using a red line.

dimension. Unlike the sequence dimension, where we pursue effective and sufficient nonlinear transformations, along the channel dimension we prefer the features between any two channels to be orthogonal. In this case, inter-channel correlation is minimized, and each channel carries complementary rather than redundant information. Therefore, we design another loss function to enforce orthogonality of layer output along the channel dimension. It aims to reduce inter-channel redundancy and improve parameter utilization.

Specifically, we let the layer output be $h_{\text{out}} \in \mathbb{R}^{B \times T \times D}$, where $d = 0..D - 1$ is channel index. For each channel, we perform $L_2$ normalization along the sequence dimension. This ensures the feature vectors lie on the unit sphere for subsequent orthogonality computation.

$$\widetilde{h}_{\text{out}} = \frac{h_{\text{out}}}{\|h_{\text{out}}\|_2 + \epsilon}, \qquad G = \widetilde{h}_{\text{out}}^{\top} \widetilde{h}_{\text{out}} \in \mathbb{R}^{D \times D}, \qquad \mathcal{L}_{\text{ortho}} = \|G - \mathbf{I}_D\|_F^2.$$

Where $\epsilon = 10^{-8}$. We construct the Gram matrix of channel features to capture their inner-product relations. Ideally, if channels are perfectly orthogonal, the Gram matrix $G$ should approach the identity matrix $\boldsymbol{I}$ whose diagonal elements are 1 and off-diagonals are 0. This indicates the Gram matrix is full-rank and positive definite, which implies channel features are linearly independent and each channel carries complementary information. To quantify the degree of deviation from orthogonality, we compute the squared Frobenius norm between $G$ and $\boldsymbol{I}$ as the orthogonality penalty.

At this point, we have designed the coherence-based redundancy loss along the sequence dimension and the orthogonality loss along the channel dimension. They achieve supervision and constraints on the model layer output features from different dimensions. And they jointly reduce the degree of redundant transformations in the model layer outputs and improve model parameter utilization.

In this study, the overall training objective combines the original cross-entropy loss for next-token prediction with the coherence-based redundancy loss and the channel-orthogonality loss. The final loss function is defined as:

$$\mathcal{L} = \mathcal{L}_{\text{ce}} + \lambda_{\text{CR}} \, \mathcal{L}_{\text{CR}} + \lambda_{\text{ortho}} \, \mathcal{L}_{\text{ortho}}.$$

Where $\lambda_{\text{CR}} = \lambda_{\text{ortho}} = 0.01$. This loss aims to guide the model to accurately predict the next token while prompting the model layers to reduce redundant transformations. And it guides the model to make full use of parameters and maintain diverse and informative representations. Subsequently, we conduct extensive pre-training experiments to validate the effectiveness of the coherence-based redundancy loss and the channel-orthogonality loss.

## 4 EXPERIMENT

We build our framework on llama3-130M. In our experiments, the pre-training hyperparameter settings follow the specifications of GPT-2 Radford et al. (2019). We train the models on an 11B token subset of The Pile. And we pre-train all models from scratch using a single A100 GPU.

Based on our analysis of experimental data, as shown in 2 (c), layers 2, 4, 6, and 8 of the model with the tree-structured residual path are more prone to containing coherence approach zero. Therefore, in our experiments, we fix the application of the coherence-based redundancy loss to layers numbered 2, 4, 6, and 8 based on the tree-structured residual path. At the same time, we apply the channel-orthogonality loss to layers numbered 3, 5, 7, 9, and 10. We first conduct ablation experiments to validate the effectiveness of the hyperparameter settings in the CR loss. As shown in 3 (a), the result demonstrates that sharpening the features before the softmax operation is necessary. By applying sharpening and enforcing an upper bound, the CR loss can function more effectively. Since we do not want the coherence to approach 0 or 1, we initially select a target value near 0.5. Furthermore, according to the coherence measurements of each layer in 2 (a), we consider it more appropriate to set the target in the range $[0.3, 0.4]$. As shown in 3 (b), the experiments confirm that when $target = 0.35$, the CR loss not only suppresses redundant output from model layers but also better guides the model to learn effective features. Additionally, using $scale_L$ reduces fluctuations in deep layers, as shown in 3 (c), and improves the model's prediction accuracy. These experiments demonstrate that the above measures can be deeply integrated with the tree-structured residual path, which enables efficient information exchange between shallow and deep layers. This enables effective regularization of the model layers even when only mild constraints are applied to the deep layers.

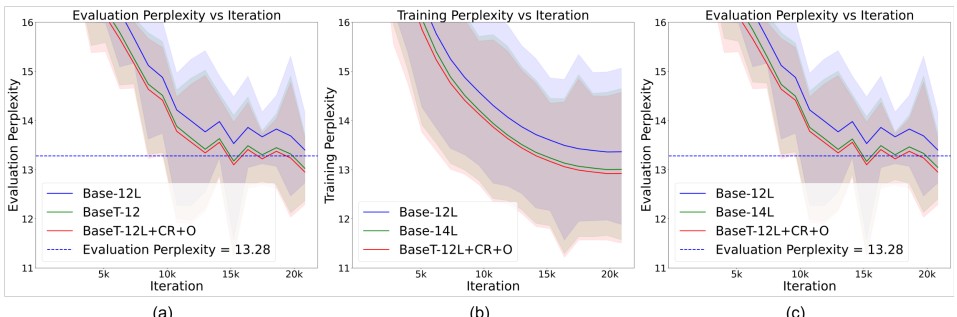

(a)  (b)  (c)

Figure 4:  Comparison of evaluation results for pre-trained models. Base-12L and Base-14L respectively denote baseline models with 12 and 14 layers. BaseT-12L denotes the 12-layer model obtained by adjusting Base-12L with the tree-structured residual path. BaseT-12L+CR+O denotes the model obtained by applying the complete regularization loss to BaseT-12L.

Compared to the 12-layer baseline, both our proposed tree-structured residual path and the redundancy regularization scheme effectively improve the parameter utilization of the model. Without increasing the number of parameters, our schemes enable models of the same parameter scale to achieve better evaluation performance, as shown in 4 (a). Ultimately, with all other training settings held constant, our schemes enable the 12-layer model to outperform the 14-layer baseline, as shown in 4 (b) and (c). According to empirical measurements, our 12-layer model achieves an evaluation perplexity that is 0.45 lower than the 12-layer baseline and 0.1 lower than the 14-layer baseline.

## 5   CONCLUSION

In this paper, we thoroughly analyze the phenomenon of redundant transformations. And we point out that the occurrence of redundant transformations is due to imperfect training paradigms. To address this issue, we establish the criteria for identifying redundant transformations and further propose the coherence-based redundancy (CR) to measure the degree of redundant transformations. Guided by the CR, we propose the tree-structured residual path and the redundancy regularization schemes to guide and constrain the middle-to-deep layers. They successfully suppress redundant outputs from model layers and effectively refine the training paradigms. Ultimately, with all other training settings held constant, our schemes enable the 12-layer model to outperform the 14-layer baseline.

## 6 REPRODUCIBILITY STATEMENT

Our results comply with the reproducibility requirements of ICLR. In this paper, we provide a complete description of the computation of the coherence-based redundancy (CR) measure. The core design of the tree-structured residual path is fully described in the paper, and the complete implementation is provided in the appendixA.2. The core formulations of the coherence-based redundancy loss and the channel-orthogonality loss are fully presented, and all implementation details are provided in the appendix A.3.

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

## A APPENDIX

### A.1 THE USE OF LARGE LANGUAGE MODELS (LLMs)

In this paper, we used large language models (LLMs) solely for text polishing, with the aim of improving fluency and readability.

## A.2 PSEUDOCODE FOR TREE-STRUCTURED RESIDUAL PATH

```
FUNCTION ForwardWithTreeResidual(
    h,                        # [B, T, D] embedding hidden states
    layers,                   # list of 12 transformer blocks
    start_pos, freqs_cis, mask, # decoding/cache inputs
    norm_1_0, norm_1_1,
    norm_2_0, norm_2_1,
    norm_4_0, norm_4_1,
    norm_6_0, norm_6_1,
    norm_8_0, norm_8_1
):
    # Layer 0
    input = h
    h0 = layers[0](input, start_pos, freqs_cis, mask)

    # Layer 1
    input = norm_1_0(h) + norm_1_1(h0)
    h1 = layers[1](input, start_pos, freqs_cis, mask)

    # Layer 2
    input = norm_2_0(h0) + norm_2_1(h1)
    h2 = layers[2](input, start_pos, freqs_cis, mask)

    # Layer 3
    input = h2
    h3 = layers[3](input, start_pos, freqs_cis, mask)

    # Layer 4
    input = norm_4_0(h0) + norm_4_1(h3)
    h4 = layers[4](input, start_pos, freqs_cis, mask)

    # Layer 5
    input = h4
    h5 = layers[5](input, start_pos, freqs_cis, mask)

    # Layer 6
    input = norm_6_0(h1) + norm_6_1(h5)
    h6 = layers[6](input, start_pos, freqs_cis, mask)

    # Layer 7
    input = h6
    h7 = layers[7](input, start_pos, freqs_cis, mask)

    # Layer 8
    input = norm_8_0(h1) + norm_8_1(h7)
    h8 = layers[8](input, start_pos, freqs_cis, mask)

    # Layer 9
    input = h8
    h9 = layers[9](input, start_pos, freqs_cis, mask)

    # Layer 10
    input = h9
    h10 = layers[10](input, start_pos, freqs_cis, mask)

    # Layer 11
    input = h10
    h11 = layers[11](input, start_pos, freqs_cis, mask)

    RETURN h11
END
```

Listing 1: Forward pass with the tree-structured residual path (12 layers).

### A.3 PSEUDOCODE FOR CR AND ORTHO LOSSES

```
FUNCTION OrthoLoss(h_out, epsilon, lambda_ortho):
    # h_out: [B, T, D]
    h_out_norm = h_out / (L2Norm(h_out, dim=1, keepdim=True) + epsilon)
    G = Transpose(h_out_norm, dim1=1, dim2=2) @ h_out_norm         # [B,
    D, D]
    I = Identity(D)
    ortho_penalty = Mean((G - I)^2)
    RETURN lambda_ortho * ortho_penalty
END

FUNCTION CRLoss(h_in, h_out, iteration, layer_num, epsilon, lambda_cr):
    # Standardize along sequence
    z_in  = (h_in  - Mean(h_in,  dim=1, keepdim=True)) / (Std(h_in,  dim
    =1, keepdim=True) + epsilon)
    z_out = (h_out - Mean(h_out, dim=1, keepdim=True)) / (Std(h_out, dim
    =1, keepdim=True) + epsilon)

    # Depth-aware clipping and scaling
    bound = 2.0 + layer_num/2
    fac_in  = maximum(MaxAbs(z_in,  dim=1, keepdim=True) / bound, 1)
    fac_out = maximum(MaxAbs(z_out, dim=1, keepdim=True) / bound, 1)
    depth_scale = sqrt(12 - layer_num)
    z_in  = z_in  * depth_scale / fac_in
    z_out = z_out * depth_scale / fac_out

    # Softmax along sequence (probability distributions)
    p_in  = Softmax(z_in,  dim=1)      # [B, T, D]
    p_out = Softmax(z_out, dim=1)      # [B, T, D]

    # ECF bases
    K = floor(T/2) + 1
    freqs = 2*pi*arange(0..K-1)/T
    t = arange(0..T-1)
    COS[k,t] = cos(freqs[k]*t)
    SIN[k,t] = sin(freqs[k]*t)

    # Channel-wise transforms: [B, D, T]
    P_in  = Permute(p_in,  [0,2,1])
    P_out = Permute(p_out, [0,2,1])

    Re_in  = Einsum("bdt,kt->bdk", P_in,  COS)
    Im_in  = Einsum("bdt,kt->bdk", P_in,  SIN)
    Re_out = Einsum("bdt,kt->bdk", P_out, COS)
    Im_out = Einsum("bdt,kt->bdk", P_out, SIN)

    Amp_in  = sqrt(Re_in^2  + Im_in^2  + epsilon) + epsilon
    Amp_out = sqrt(Re_out^2 + Im_out^2 + epsilon) + epsilon

    # Energy-ratio penalty (early iterations)
    Power_in  = Sum(Amp_in,  over=k)
    Power_out = Sum(Amp_out, over=k)
    power_ratio = Power_out / (Power_in + epsilon)
    threshold = 2.0
    energy_penalty = ReLU( (log(power_ratio))^2 - (log(threshold))^2 )  #
     [B, D]

    # Magnitude-squared coherence (batch-mean)
    C_in  = Re_in + i*Im_in
    C_out = Re_out + i*Im_out
    S_xy = Mean_B( C_in * Conj(C_out) )                # [D, K]
    S_xx = Mean_B( |C_in|^2 ) + epsilon                # [D, K]
    S_yy = Mean_B( |C_out|^2 ) + epsilon               # [D, K]
    coherence = |S_xy|^2 / (S_xx * S_yy + epsilon)     # [D, K] in [0,1]
```

```
coherence = Unsqueeze(coherence, dim=0)              # [1, D, K] -> [B,
 D, K]

    # coherence targets
    target = 0.35

    similarity_penalty = (coherence - target)^2
    similarity_penalty = similarity_penalty * ((12 - layer_num)/2) * 0.2
    similarity_penalty = Mean(similarity_penalty, over=k)          # [B, D]

    IF iteration <= 800:
        penalty = similarity_penalty + energy_penalty  # warm-up stage
    ELSE:
        penalty = similarity_penalty

    CR_loss = lambda_cr * Mean(penalty)
    RETURN CR_loss
END

FUNCTION TrainStep(model, batch, iteration, epsilon=1e-8, lambda_cr=0.01,
     lambda_ortho=0.01, ce_weight=1.0):
    logits, {attention_sub_layer_inputs[h_in_l],
    attention_sub_layer_outputs[h_out_l]} = model.forward_with_features(
    batch)
    loss_ce = CrossEntropyLoss(logits, batch.targets)

    loss_cr = 0
    loss_ortho = 0

    FOR layer_num IN 1..L:
        h_in  = attention_sub_layer_inputs[layer_num]
        h_out = attention_sub_layer_outputs[layer_num]

        IF layer_num IN {2, 4, 6, 8}:
            loss_cr += CRLoss(h_in, h_out, iteration, layer_num, epsilon,
     lambda_cr=0.01)

        IF layer_num IN {3, 5, 7, 9, 10}:
            loss_ortho += OrthoLoss(h_out, epsilon, lambda_ortho=0.01)
    END FOR

    total_loss = ce_weight*loss_ce + loss_cr + loss_ortho
    BACKPROP(total_loss)
    UPDATE(optimizer)

    RETURN {total_loss, loss_ce, loss_cr, loss_ortho}
END
```

Listing 2: CR loss, Ortho loss, and training step.

