# OpenReview forum: "CR-Guided Transformers: Coherence-Based Redundancy Identification and Regularization"
_ICLR.cc/2026/Conference — Submitted to ICLR 2026_

### Official Review · Reviewer_UawZ · 2025-10-30

**Soundness:** 2
**Presentation:** 2
**Contribution:** 2
**Rating:** 0
**Confidence:** 4

**Summary:**

The paper tries to address the problem that layers in mid-to-deep layers don't do too much anymore. The motivation is to make better use of the model parameters.

Some additional residual connections are introduced: tree-structured residual path, which effectively reduces the total depth, to more directly connect to mid-to-deep layers.

A new kind of regularization is introduced, based on the coherence between input and output of a layer. The coherence is measured in a differential way, and you specify a wanted target coherence, and then the squared difference between target coherence and actual coherence is added as an auxiliary loss for better regularization.

Llama3-130M is used as the baseline for experiments, with 12 layers and 14 layers. It is trained on a subset of 11B token of The Pile. Evaluation is done via perplexity on some eval set.

The tree-structured residual path seems to contribute mostly to the performance, while the additional regularization only gives small improvements. It seems that a 12 layer model with that extra tree-structured residual path and the regularization slightly outperforms a 14 layer baseline model.

**Strengths:**

* New tree-structured residual path.
* New regularization method.
* Some analysis of coherence of different layers.

**Weaknesses:**

* The presentation has many issues (see questions/comments below).
* Training speed / overhead for the tree-structured residual path not measured / reported.
* Training speed / overhead for the additional regularization not measured / reported.
* The experiments are way too limited. This is by far the biggest issue.
  * 20k iterations is way too little training.
  * Models are way too small.
  * Model size variations / scaling law studies missing.
  * Other (more relevant) model types not studied, e.g. Transformer++.
  * Evaluation not done properly, should not just measure perplexity but also other benchmarks.
  * Ablations way too limited.
* No code published.

**Questions:**

"As shown in 1 (a) and 2 (a)" and many more such examples: It misses some "Figure", i.e. "as shown in Figure 1 (a)".

Add equation numbers. That makes it easier to argue about.

In the def of L_CR, Coherence is not defined. Before, you define Coherence as a function of k, but here, it is not a function of k. What is it?

Also, in the def of L_CR, you should make proper use of Latex. Use `\operatorname` or so.

In the def of L_CR, this is depending on the layer index L? (Btw, capital L is weird for layer index. Capital L is usually the num of layers.) But then, in the final loss, you also use L_CR, but there is no L anymore? So this does not fit together.

Looking at Figure 4, it looks like most of the improvements are because of the tree-structured residual path, while the improvements from the new regularization is minimal?

Figure 4 (a) and Figure 4 (c), why to have them separate and not in a single plot? They look like they should be together? Also, they look extremely similar anyway...

What is the training speed overhead for the tree-structured residual path? Or is there any?

What is the training speed overhead for the additional regularization? I assume this will be some overhead?

20k iterations is way too little training. Train much longer. Also, train on larger models. Also, train on multiple different sizes, to better see the overall trends. Also, train with other model variants, e.g. Transformer++.

Also, don't just show perplexity. Do evaluation on some other benchmarks.

There could be more studies on variants of ways how to measure the coherence. There are many possible other ways how to measure it, e.g. any kind of similarity measure.

Figure 1 (c) is a bit unclear. There is an outgoing arrow for E, 0 and 1. Where does it go? There is an incoming error for 0, 1 and 2. Where does it come from? Just connect all the arrows and don't let me guess on this. Also, I guess you also have the normal residual connections? Just make them all explicit. When you add them up, also make that explicit, by adding an "+" node in the graph.

The exact definition of the model with the tree-structured residual path is not clear for me. Despite Figure 1 (c) (which itself is not totally clear, see above), also write down the exact mathematical definition.

No code is published?

**Details Of Ethics Concerns:**

.

---

### Official Review · Reviewer_Y61Q · 2025-10-30

**Soundness:** 1
**Presentation:** 1
**Contribution:** 1
**Rating:** 0
**Confidence:** 4

**Summary:**

This paper proposes a new transformer architecture using some newly proposed coherence-based redundancy metrics. The idea could compress the 14-layer model to 12 layers and has only been tested on one 130M LLaMA model. The writing is very poor, and the experiments are incomplete. Even the citation format is bad.

This paper just doesn't match my expectations for an ICLR paper (I actually asked several colleagues and they agree with me). Is this a test from the OpenReview foundation to let human reviewers review AI-generated papers?

**Strengths:**

The only strength I could think is the proposed idea might work. But the current experiment is insufficient to prove this.

**Weaknesses:**

1. Please use \citep{} instead of \cite{}. The citations of the whole paper are mixed with the main content.

2. The logic is bad. At the very first in the abstract, there are words like "redundant transformations", "nonlinear transformations", "Coherence-based Redundancy", which I just have no idea what they are.

3. The paper seems very redundant itself and is badly structured. The introduction takes up space until half of page 3, but there isn't any section discussion related works?

4. Figure 1 is too simple to convey the research idea. I can understand the proposed tree-structure paths, but cannot get how the training and inference are conducted, and if there are any shared parameters.

5. I did not examine the method section in detail due to time constraints, but the experimental evaluation appears highly incomplete. Using only a 130M LLaMA model and comparing perplexity against the base model itself is insufficient. A more comprehensive set of datasets, tasks, and baseline comparisons is necessary to meet the standards expected for ICLR publications.

With my full respect, I doubt this paper is AI-generated. It is possible that I just don't have enough background to understand this paper (I know transformers and common ML architectures very well, maybe the coherence-based redundancy topic requires a lot specific knowledge that I don't have yet).

**Questions:**

N/A

If there is any human author of this paper, could you let me know?

I don't mean to be harsh, but not checking clear formatting issues before submission is already disrespectful to reviewers.

---

### Official Review · Reviewer_5Rbt · 2025-10-31

**Soundness:** 3
**Presentation:** 2
**Contribution:** 3
**Rating:** 4
**Confidence:** 3

**Summary:**

The paper investigates the phenomenon seen in many transformer models that layer features often have a high cosine similarity, indicating redundancy and inefficient training. A new measure of redundancy - coherence - is proposed, with the claim that it does a better job in indicating redundancy. A new architecture and training protocol is proposed which uses the coherence measure to control and reduce layer redundancy, resulting in enhanced performance with lower computation costs.

**Strengths:**

The paper identifies an important shortcoming of current LLMs that of layer redundancy, and proposes a novel measure for indicating the redundancy and a novel training technique and architecture to remedy this problem.
Non-trivial experiments show that the proposed technique can enhance accuracy of an LLM while reducing computation.

**Weaknesses:**

The proposed approach may be good, but it is not explained very well in the paper. It is not clear why the proposed coherence measure is any better that the cosine similarity for the purpose of measuring and reducing feature redundancy.

line 117: "We elucidate the criteria for identifying redundant transformations." This is also mentioned in the abstract and in the conclusion. But nowhere in the paper is a criteria for identifying redundant transformations clearly defined. Since this is one of the stated contributions it should be very clearly defined in the text somewhere.

line 140: This sentence doesn't make sense, and should be rewritten: "It is very high that the average cosine similarity of some middle-to-deep transformer layers, as shown in 1 (a)."

The notion of redundancy as being indicated by high cosine similarity is imprecise. Even a high (but not 1.0) cosine similarity indicates some difference between the features. It may be that this small difference is useful, or even critical, in performing the task, to represent small but crucial bits of detail distinguishing one class from another. The paper needs to be more precise in specifying what cosine similarity level constitutes an irredeemable redundancy.

line 222: "Although prior research Men et al. (2024) can simply obtain correlations between features by calculating cosine similarity." This sentence is grammatically incorrect and should be rewritten.

line 238: " Combined with the criteria for redundant transformations...". What IS the criteria for redundant transformations?

line 278: "It [coherence] can reveal statistical structural differences after nonlinear transformations more effectively than cosine similarity, which relies solely on directional information. As shown in 1 (a) and 2 (a), in the baseline model the input–output coherence and the cosine similarity exhibit the same trend from 1 to 10 attention sub-layers".  The second sentence contradicts the first, as it implies that coherence and cosine similarity give the same result. Why do we need to complicate matters using coherence? Why not just use cosine similarity to measure and control redundancy?

**Questions:**

It seems that what is not mentioned in this paper, which should be I think, is that layer redundancy also means that training time is being wasted. With large models taking very long times to train, resulting in high costs and resource usage, reducing training time or wasting less training time is an important consideration. I suggest the authors add this to the motivation for their method, as opposed to say pruning which reduces network size but does not reduce wasted training time.
Of course, one counter-argument is that the issue is not inefficient training, but perhaps the redundancy is due to insufficient training. Would continued training, with new or augmented data force the network to reduce layer redundancy?

Line 77: what proof or evidence is there that distributions are more easily separated in the complex plane?

On line 156: "If the residual branch gradient term is too small, the derivative of this layer tends to approach the identity matrix". But what evidence is there that this gradient term is actually too small? How small is too small?

Line 183: "this paper is devoted to accurately identifying redundant transformations". How is accuracy defined here? Is there some ground truth for redundant transformations? If not, how can one talk about accuracy in this context?

Line 237: "And frequency-domain coherence is more analytical." What is meant by "analytical" here? With regard to complex signals, analytic refers to complex-valued signals with no negative frequency components.

Line 286: "Combined with the criteria for redundant transformations, it can be inferred as follows. When coherence approaches 0, it indicates invalid feature transformation. And when coherence approaches 1, it indicates insufficient nonlinear transformation between features.". Where is the proof of these statements? Why does a coherence of 0 indicate an invalid feature transformation? Why does a coherence of 1 indicate insufficient nonlinear transformation between features? These statements may be true, but there is no proof given.

---

### Official Review · Reviewer_6rSa · 2025-11-04

**Soundness:** 2
**Presentation:** 2
**Contribution:** 2
**Rating:** 2
**Confidence:** 4

**Summary:**

The paper studies redundancy in Transformer layers. They formalize two intuitive criteria:
1. if the input/output have approximately linear correlation
2. the input/output have almost zero correlation

where if one of the above is met, it means that the layer is redundant.

Instead of using cosine similarity as previous work they propose to use characteristic functions and analysis in the frequency domain. The authors also propose to change the residual connection structure in the Transformer based on a tree structure to better model longer range dependencies between layers and redundancy regularization.

My main concern about this paper is insufficient experimental validation and comparison with alternative pruning techniques.

**Strengths:**

The paper tackles an important problem of improving the information efficiency of the Transformer architecture.

**Weaknesses:**

As mentioned above, the authors main experimental validation experiment is in Figure 4. The evaluation is based on perplexity and the authors' approach is only compared with a baseline model. It is essential that the author's also evaluate on downstream tasks/metrics and compare with other alternative techniques for layer pruning e.g. see  Hoefler et al. (https://arxiv.org/abs/2102.00554), Ma et al. (http://arxiv.org/abs/2305.11627), Men et al. (https://arxiv.org/abs/2403.03853) and the references therin.

**Questions:**

See weaknesses above.

---

### Meta-Review · Area_Chair_daH4 · 2026-01-06

**Summary:**

All reviewers agree that the current draft does not explain the proposed idea very well, has many issues on the presentation, and lacks experiments.

Specifically, reviewers raised concerns about:
1) the evaluation relies almost exclusively on perplexity. Reviewers demand evaluation on downstream tasks, standard benchmarks, and a wider variety of datasets;
2) the paper fails to compare the proposed approach against existing layer-pruning or redundancy-reduction techniques;
3) the paper fails to justify why the proposed coherence measure is superior to standard cosine similarity;
4) also, the overall quality of the manuscript was heavily criticized.

**Reviewer Concerns:**

No rebuttal, so all the concerns have not been resolved.

**Reviewer Scores:**

The reviewers will not change their scores as no rebuttal is posted.

---

### Decision · Program_Chairs · 2026-01-26

Reject